# A Molecular Tool for Rapid Detection and Traceability of *Cyclospora cayetanensis* in Fresh Berries and Berry Farm Soils

**DOI:** 10.3390/foods9030261

**Published:** 2020-03-02

**Authors:** Carolina N. Resendiz-Nava, Guadalupe E. Orozco-Mosqueda, Edmundo M. Mercado-Silva, Susana Flores-Robles, Hilda V. Silva-Rojas, Gerardo M. Nava

**Affiliations:** 1Facultad de Química, Universidad Autónoma de Querétaro, Cerro de las Campanas S/N, Querétaro 76010, Mexico; carolina.resendiz.90@gmail.com (C.N.R.-N.); mercado501120@gmail.com (E.M.M.-S.); juana.susana.flores@uaq.mx (S.F.-R.); 2Hospital Infantil de Morelia Eva Samano de López Mateos, Servicio de Salud de Michoacán, Interior Bosque Cuauhtémoc S/N, Michoacán 58020, Mexico; erendira4@hotmail.com; 3Posgrado en Recursos Genéticos y Productividad, Producción de Semillas, Colegio de Postgraduados, Km. 36.5 Carretera Mexico-Texcoco, Estado de Mexico 56230, Mexico; hsilva@colpos.mx

**Keywords:** *Cyclospora cayetanensis*, PCR, Mexico, berries, soil

## Abstract

Due to recent outbreaks of cyclosporiasis associated with consumption of fresh berries, producers are demanding modern microbiological tools for the rapid and accurate identification of the human pathogen *Cyclospora cayetanensis* in berries and environmental samples. The aim of the present work was to develop a molecular tool based on a PCR approach for the rapid and accurate detection of *C. cayetanensis*. A nested PCR assay was validated for the amplification of a 294 bp size region of the *18S rRNA* gene from *C. cayetanensis*. The limit of detection for the nested PCR assay was validated using 48 berry samples spiked with ~0, 10, 100, and 1000 oocyst per gram of sample. With this assay, it was possible to detect as few as 1 oocyst per gram of berry, in a 50 g sample. Sanger DNA sequencing and phylogenetic analysis were carried out to confirm the presence of *C. cayetanensis* in berry (n = 17) and soil (n = 5) samples. The phylogenetic analysis revealed that the *C. cayetanensis* sequences obtained from Mexico clustered within a group recovered from China, Peru, Guatemala-Haiti, and Japan. The PCR protocol designed in the present study could be an important tool for the rapid and accurate detection of this human pathogen in environmental and food samples.

## 1. Introduction

The consumption of raspberries (*Rubus idaeus*), blackberries (*Rubus* sp.), and blueberries (*Vaccinium corymbosum*) has increased worldwide in recent years because these fruits are considered an important source of antioxidant compounds [1,2]. Unfortunately, consumption of berries is associated with a risk of acquiring foodborne parasites, such as *Cyclospora cayetanensis* [1,3]. For example, in the USA and Canada, 4250 cyclosporiasis cases have been linked to consumption of raspberries, blackberries, and strawberries [1]. In the UK, 43 cases of cyclosporiasis were linked to the consumption of fresh strawberries and raspberries [4]. At the farm level, the presence of *C. cayetanensis* in berries is directly associated with the presence of the parasite in soil [5,6,7]. Thus, it is fundamental that producers monitor the presence of this pathogen on farms and packing facilities.

Conventionally, detection of *C. cayetanensis* in clinical and environmental samples is based on identification of oocysts by microscopy, following modified acid fast staining or by autofluorescence under ultraviolet (UV) light [8]. However, this technique is time-consuming, non-specific, and lacks sensitivity [9]. To overcome these issues, molecular methods have been developed to detect *C. cayetanensis* in clinical and environmental samples [10,11,12]; however, the food producing industry requires a molecular method able to detect a low oocyst concentration (40–1500 oocyst per gram) as found in food and environmental samples [13]. Also, it is important to have a tool available to perform molecular traceability and identification of contamination sources. Thus, the objective of the present study was to develop and validate a highly sensitive and specific PCR assay for the rapid and accurate detection of *C. cayetanensis*, as well as its molecular traceability in fresh berries and farm soils.

## 2. Materials and Methods

### 2.1. DNA Extractions

Oocysts of *C. cayetanensis* (preserved in 2.5% potassium dichromate solution) obtained from a laboratory strain collection and environmental (fruit and soil) samples were subjected to DNA extraction using the ZymoBIOMICS DNA Kit (Zymo Research, Irvine, CA, USA) following manufacturer’s instructions. Purified DNA was diluted to reach a concentration of 1 ng/µL and stored at −20 °C.

### 2.2. Nested PCR Assay

To identify the most effective conditions for nested PCR, gradient PCR amplifications were performed using different annealing temperatures (ranging from 48 °C to 75 °C) for each primer set. Various rounds of nested PCR reactions were also carried out using *C. cayetenensis* purified DNA and DNA from PCR amplification. Overall, the first round of PCR amplification using primer pair CYCF1E (5′-TACCCAATGAAAACAGTTT-3′) and CYCR2B (5′-CAGGAGAAGCCAAGGTAGG-3′), generated a ~630 bp amplicon [10]; this primer pair amplifies a segment of the *18S rRNA* gene found in different members of the *Eimeriidae* family [14]. Each PCR reaction (17 µL) contained: 3.4 µL of 5X Phire Hot Start II DNA Polymerase (Thermo Scientific, Waltham, MA, USA) reaction buffer, 0.34 µL of dNTP solution mix (10 mM) (Thermo Scientific.), 0.34 µL of Phire Hot Start II DNA Polymerase, 0.68 µL of each primer (1.0 µM), 2 µL of bovine serum albumin (Bioline, London, UK), and 3.0 ng of DNA extracted from oocysts. The amplification program consisted of 1 min at 95 °C followed by 35 cycles of denaturing at 95 °C for 30 s, annealing at 53.6 °C for 30 s, an extension at 72 °C for 30 s. and a final extension for 2 min at 72 °C. Genomic DNA (3.0 ng) obtained from *C. cayetanensis* oocysts was used for positive control reactions. The second round of PCR amplification was carried out using primer pair CC719 (5′-GTAGCCTTCCGCGCTTCG-3′) and CRP999 (5′-CGTCTTCAAACCCCCTACTGTCG-3′), which generates a ~298 bp amplicon [15]. The specificity of this primer pair has been previously validated against other *Cyclospora* species and genera of the *Eimeriidae* family [15,16]. The PCR reaction was performed as described above, using an annealing temperature of 66.5 °C and 2.0 µL of 1:100 diluted PCR products from the first reaction. All PCR products were subjected to electrophoresis using 1.5% (wt/vol) agarose-TBE (89 mM Tris-borate, 2 mM EDTA) gels, stained with an *ethidium bromide solution*(10 µg/35 mL gel) (Bio-Rad, Hercules, CA, USA). Specificity of the nested PCR assay was corroborated by Sanger sequencing and phylogenetic analysis.

### 2.3. Sensitivity of the Nested PCR Assay

To establish the limit of detection of the PCR assay, 50 g of fresh blueberries were spiked (applied as droplets) with approximately 1, 10, 100, and 1000 *C. cayetanensis* oocysts per gram of sample, using an inoculation solution (10 oocysts/µL). Briefly, approximately 3 mL of oocysts in 2.5% potassium dichromate solution were washed three times with 10 mL sterile distilled water by centrifugation (4500 rpm) at room temperature during 10 min to remove potassium dichromate. Washed cells were resuspended in 1 mL of sterile saline solution. Concentration of oocysts was estimated by counting three replicates using a hemocytometer. To evaluate reproducibility, experiments were carried out in triplicate as follows. Spiked blueberries were kept overnight at 4 °C to promote oocyst adherence to the fruit [17]. Then, samples were rinsed manually (~5 min) with 50 mL of sterile distilled water using a sterile plastic bag; washes were transferred to a conical tube and centrifuged for 10 min at 10,000 rpm [17] to recover a sample pellet. Approximately, 250 mg (between 80–100%) of the pellet was used for DNA extraction, and the genomic DNA was subjected to the nested PCR assay as described above.

### 2.4. Detection of C. cayetanensis in Fresh Berries and Farm Soils

To validate the usefulness of the nested PCR assay under field conditions, samples of blueberries (n = 6), raspberries (n = 6), blackberries (n = 11), and farm soils (n = 5) were obtained from commercial farms located in central Mexico. Each fruit sample consisted of 500 g of berries, which were rinsed (as described above) with 100 mL of sterile distilled water and subjected to DNA extraction (as described above). Soil samples consisted of ~50 g of soil collected at the base of the plants; ~250 mg of each soil sample was used for the DNA extraction. Purified DNA was subjected to the nested PCR protocol as described above. PCR products were subjected to Sanger sequencing and phylogenetic analysis. Each fragment was sequenced at least twice to eliminate sequencing errors.

### 2.5. Molecular Traceability of C. cayetanensis

The potential use of the *18S rRNA* fragments, obtained with the nested PCR, was evaluated for studies of molecular traceability of pathogen origin. To accomplish this goal, a genomic survey at the GenBank was carried out using BLAST search tool [18]. Fifty-five sequences from *C. cayetanensis 18S rRNA* genes were found and retrieved from the GenBank. This set contained sequences from ten different countries (Table 1). Sequences were aligned, manually trimmed, and edited using MEGA 6 software [19]. Nucleotide alignments were subjected to phylogenetic analysis using the Maximum Likelihood [20], Maximum Parsimony [21], and Neighbor Joining [22] methods, *Jukes-Cantor evolutionary model* [23], *and* bootstrap analysis after 1000 replicates. The *18S rRNA* gene sequences obtained in the present study were deposited in the GenBank under accession numbers: MK332310 to MK332315.

## 3. Results and Discussion

### 3.1. Nested PCR Standardization

To improve sensitivity in PCR-based detection of *C. cayetanensis*, two strategies were implemented; first, a nested PCR approach was designed because this pathogen is found at low concentrations in fruits and environmental samples [13]. Second, PCR assays were designed to amplify the *18S rRNA* locus because multi-copy genes increase PCR sensitivity [30]. The first round of amplification using primer pair CYCF1E and CYCR2B generated specific and strong PCR signals when 3.0 ng of DNA template per reaction and an annealing temperature of 53.6 °C were used. For the second round of amplification, primer pair CC719 and CRP999 generated specific and strong PCR signals when the first-round PCR products were diluted 1:100 and an annealing temperature of 66.5 °C was used (Figure 1). Specificity of the nested PCR assay was corroborated by Sanger sequencing and phylogenetic analysis. These analyses revealed that nested PCR fragments obtained with the present protocol share high genetic similarity (~98.0% identity) with *18S rRNA* genes from *C. cayetanensis* archived in the GenBank (accession numbers: EU861001.1, KY770759.1 and GU557063.1), confirming the specificity of the nested PCR assay. Importantly, these primer sets have been included in a new U.S. Food and Drug Administration method developed for detection of *C. cayetanensis* on cilantro and raspberries [31].

### 3.2. Sensitivity of the Nested PCR Assay

To determine the sensitivity of the nested PCR protocol, blueberry samples were spiked with approximately 1, 10, 100, and 1000 *C. cayetanensis* oocysts per gram of sample. It was observed that this PCR assay can detect as few as 1 oocyst per gram of sample; i.e., 50 oocysts in 50 g of sample (Figure 2). This level of sensitivity could be attributed to the inclusion of bovine serum albumin in each PCR reaction to overcome the effects of PCR-inhibitory substances found in environmental samples [32]. Thus, this nested PCR protocol could be a useful tool for the rapid and accurate detection of *C. cayetanensis* in environmental samples where oocyst concentrations are considered low; for example, 10^2^ to 10^3^ per gram of produce [13], 10^3^ per liter of river water [6], 10^1^ per liter of tap water [11], and 10^2^ to 10^5^ per gram of stool [33]. Because an infective dose of *C. cayetanensis* has been estimated between 10^1^ and 10^2^ oocysts [34,35], this PCR assay is sensitive enough to detect these levels of contamination in berries and environmental samples. Other studies have also accomplished low levels of detection using nested PCR assays. For example, in fresh basil, lettuce, and raspberries it was possible to detect from 1 to 10 *C. cayetanensis* oocysts per gram of product [36,37]. Taken together, these results reveal that nested PCR assays are a sensitive and effective tool for the detection of *C. cayetanensis* in fresh produce.

### 3.3. Detection of C. cayetanensis in Berries and Farm Soils

To validate the effectiveness of the nested PCR assay in detecting *C. cayetanensis* in environmental samples, fresh berries and farm soil were collected, processed, and subjected to the PCR protocol. The presence of *C. cayetanensis* was detected in 16.6% (1/6), 36.4% (4/11) and 20.0% (1/5) of blueberry, blackberry, and farm soil samples, respectively (Figure 3). These results were confirmed by Sanger sequencing and phylogenetic analysis. All seven PCR products obtained from fresh berries and soil samples share high genetic similarity (~ 99.9%) with *18S rRNA* genes from *C. cayetanensis* archived in the GenBank (accession numbers: EU861001.1, KY770759.1 and GU557063.1). To the best of our knowledge, these sequences represent the first evidence of *C. cayetanensis 18S rRNA* PCR products obtained from environmental samples. Altogether, these results confirmed the high specificity of the nested PCR assay for the detection of *C. cayetanensis* in environmental samples, such as fresh berries and farm soil.

Importantly, the present study revealed the presence of *C. cayetanensis* in samples (berries and soil) collected from commercial farms, suggesting that these products could represent a potential threat to human health. Comparable results have been reported in South Korea where 2.3% of blueberries were positive for *C. cayetanensis* [38]. These results are of relevance because consumption of fresh berries [1,3,4] and contact with contaminated soil [5,6,7] have been linked to numerous *C. cayetanensis* outbreaks in humans. Together, these data highlight the importance of performing molecular surveillance for the rapid and opportune detection of this human pathogen.

### 3.4. Molecular Traceability of C. cayetanensis

An important and current challenge in food microbiology is to rapidly identify sources or origin of contamination. To this end, genomic analyses of pathogens are an important tool to accomplish this goal [39]; unfortunately, genomic information from *C. cayetanensis* is very limited, thus it is essential to generate this molecular evidence. In the present study, the use of the *18S rRNA* locus is proposed not only for rapid and accurate detection of *C. cayetanensis*, but also, as a potential tool for molecular traceability. To perform this analysis, fifty-five *C. cayetanensis 18S rRNA* gene sequences were found and retrieved from the GenBank database. These nucleotide sequences were obtained from countries such as China, Guatemala, Iran, Japan, Korea, Mexico, Nepal, Peru, Poland, and Singapore [24,25,26,27,28,40]. Six additional *18S rRNA* gene sequences (accession numbers MK332310 to MK332315) from Mexico, obtained in the present study, were included in the analysis. Phylogenetic analysis of these fifty-five gene fragments (position 760 to 986, *C. cayetanensis* AF111183.1 numbering) revealed the presence of seven different genotypes **(**Figure 4).

Genotype I combined *18S rRNA* gene sequences from Japan and Mexico. Interestingly, information available at the GenBank indicates that sequences AB368541.1 and AB368542.1 were obtained from an infected Japanese patient who traveled to Mexico, and sequence MK332314 was recovered from an environmental sample from Mexico, indicating that genotype I found in Japan is also prevalent in Mexico. This result underlines the potential use of this molecular tool for identification of pathogen origin source. Genotype II and III are integrated by *C. cayetanensis* genetic variants found in Iran and China, respectively. Genotype IV is integrated by sequences from China. Genotype V is broadly distributed and integrates a variant found in China, Guatemala, Japan, Korea, Mexico, Nepal, Peru, Poland, and Singapore. Genotype VI and VII are integrated by variants found in China. Overall, these results indicate that in China and Mexico at least five and two different genotypes, respectively; are prevalent (Figure 4). Moreover, phylogenetic analyses of *18S rRNA* genes revealed the presence of seven new polymorphic sites at positions 816 (T/G), 823 (T/C), 824 (T/C), 852 (G/A), 873 (G/A), 940 (G/C), and 958 (T/A) (*C. cayetanensis* AF111183.1 numbering) (Figure 5). These results provide preliminary information regarding molecular diversity of *C. cayetanensis* causing disease in humans. Very recently, other studies have proposed alternative loci for *C. cayetanensis* source tracking; for example; the use of whole-genome sequence analysis allowed the identification of multiple genomic regions enriched in single-nucleotide-polymorphisms, that have been used as markers to differentiate, to some extent, source of origin [41,42,43]. However, implementation of these molecular markers requires the use of multilocus sequence typing protocols (*i.e.,* DNA Sanger sequencing of multiple genes), which remain relatively expensive and laborious in many world regions. Moreover, these protocols have been implemented only in clinical samples [41,42,43] where concentration of oocysts is elevated [41].

## 4. Conclusions

Due to the current cyclosporiasis outbreaks associated with the consumption of fresh raspberries, blueberries, and blackberries, producers require a rapid and effective microbiological method for effective detection of this human pathogen. The PCR assays described in the present study showed high sensitivity and specificity for the rapid and accurate detection of low number of cells (1 oocyst per g of blueberry sample). The use of this molecular approach could provide additional nucleotide sequences from *C. cayetanensis 18S rRNA* gene, from fresh produce and environmental samples, to increase our knowledge about the diversity and distribution of this pathogen.

## Figures and Tables

**Figure 1 foods-09-00261-f001:**
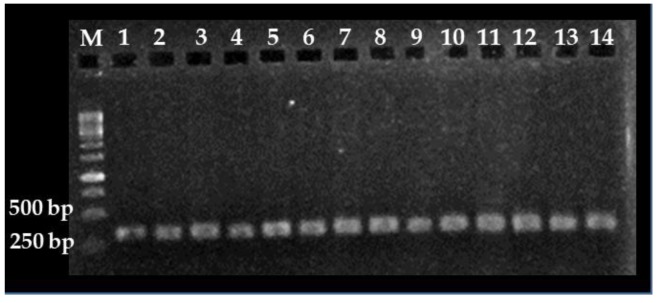
*Cyclospora cayetanensis 18S rRNA* DNA fragments obtained with the nested PCR assays. Lanes 1 to 14: DNA (3 ng) extracted from *C. cayetanensis* oocyst. M: molecular weight marker.

**Figure 2 foods-09-00261-f002:**
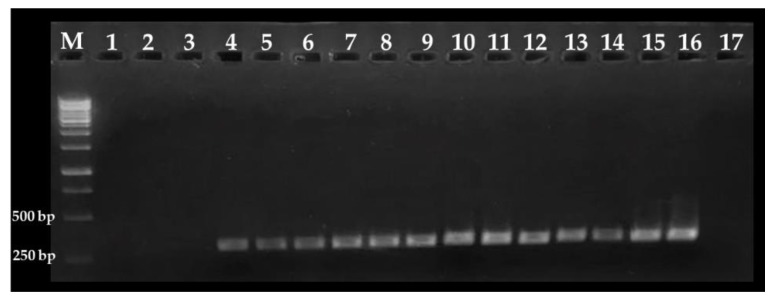
Detection limit of the *Cyclospora cayetanensis* nested PCR assay. Lanes 1 to 3: non-spiked blueberries; lanes 4 to 6: samples spiked with ~1 oocyst per gram of sample; lanes 7 to 9: samples spiked with ~10 oocysts per gram of sample; lanes 10 to 12: samples spiked with ~100 oocysts per gram of sample, and lanes 13 to 15: samples spiked with ~1000 oocysts per gram of sample. Lane 16: *C. cayetanensis* DNA as positive control and lane 17: water PCR grade as negative control. Fifty grams of blueberries were used for each sample. M: molecular weight marker.

**Figure 3 foods-09-00261-f003:**
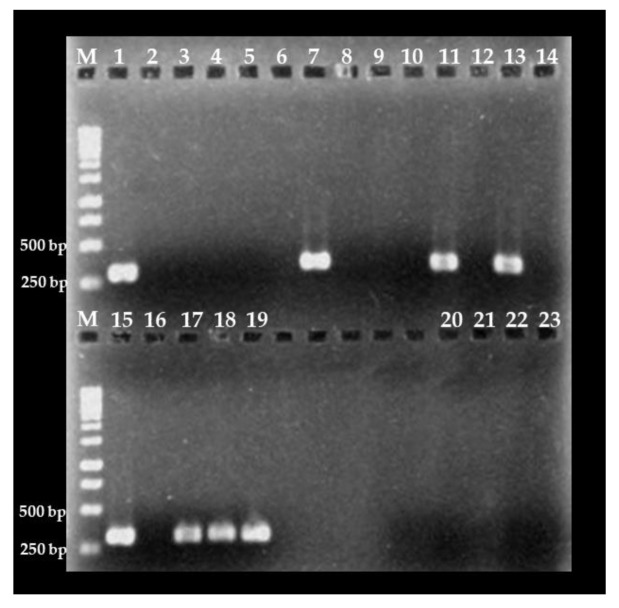
Detection of *Cyclospora cayetanensis* in berry and soil samples from Mexican farms. Lanes 1 to 3: blueberry; lanes 4 to 6: raspberry; lanes 7 to 14: blackberry samples; and lanes 15 and 16: soil samples. Lanes 17 to 19: *C. cayetanensis* DNA as positive control and lanes 20 to 23: water PCR grade as negative control. M: molecular weight marker.

**Figure 4 foods-09-00261-f004:**
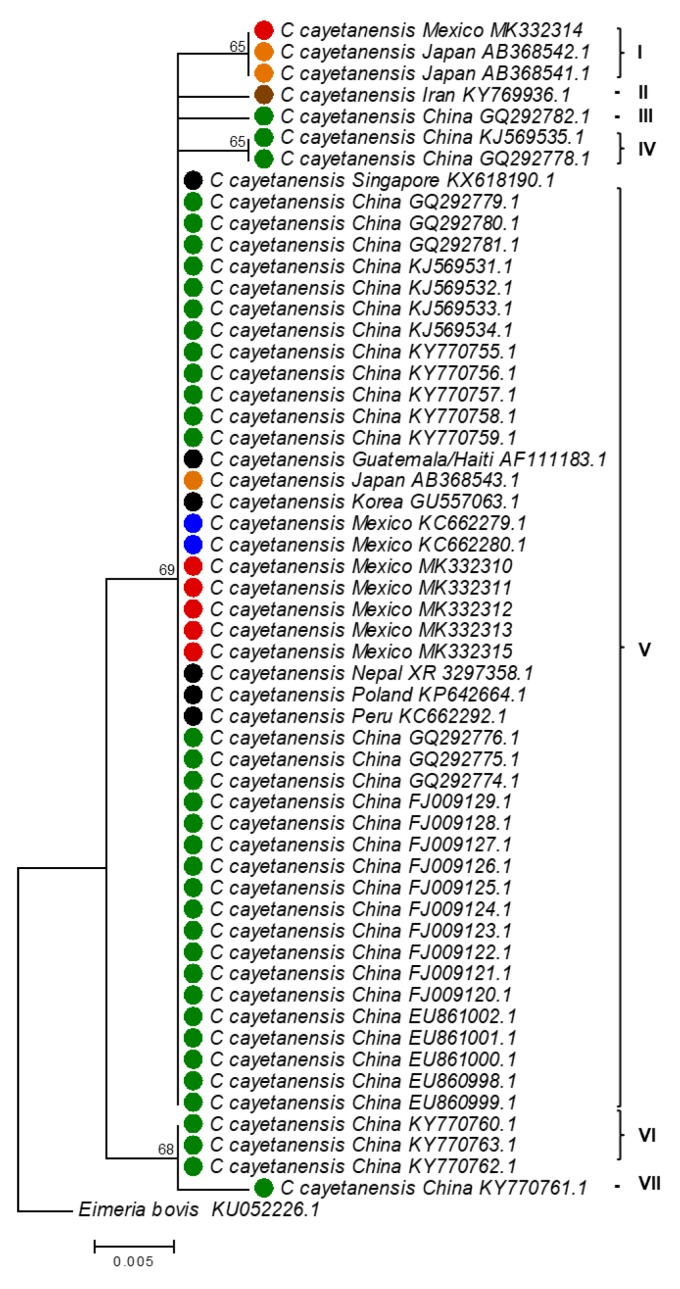
Phylogenetic analysis of *Cyclospora cayetanensis 18S rRNA* genes. Red circles: sequences from Mexico (present study); orange circles: sequences from Japan; brown circle: sequence from Iran; black circles: sequences from Singapore, Korea, Nepal, Guatemala/Haiti, Poland, and Peru; green circles: sequences from China. Roman numbers (I–VII) depict different genotypes identified in the present study. Countries of origin and accession numbers are provided for each sequence.

**Figure 5 foods-09-00261-f005:**
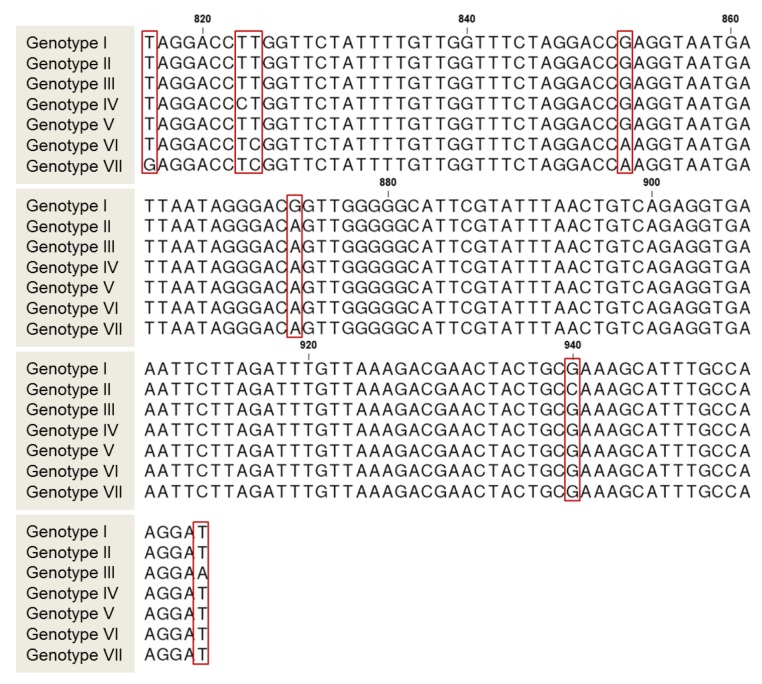
Analysis of polymorphic sites in the *Cyclospora cayetanensis 18S rRNA* gene. Red boxes enclosed new polymorphic sites identified in the present analyses. Nucleotide positions are based on *C. cayetanensis* AF111183.1 numbering.

**Table 1 foods-09-00261-t001:** Accession numbers of the downloaded sequences from the GenBank database used for the phylogenetic analysis.

Accession Numbers	Origin	Reference
AB368541–AB368543	Japan	Unpublished
AF111183	Guatemala-Haiti	[24]
EU860998–EU861002	China	[25]
FJ009120–FJ009129	China	[25]
GQ292774–GQ292776	China	[25]
GQ292778–GQ292782	China	[25]
GU557063	Korea	[26]
KC662279–KC662280	Mexico	[27]
KC662292	Peru	[27]
KJ569531–KJ569535	China	Unpublished
KP642664	Poland	[28]
KY769936	Iran	Unpublished
KX618190	Singapore	[26]
KY770755–KY770763	China	[29]
XR_003297358	Nepal	Unpublished

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
