# Peer review of "A Molecular Tool for Rapid Detection and Traceability of Cyclospora cayetanensis in Fresh Berries and Berry Farm Soils"

_foods, 2020, doi:10.3390/foods9030261_

Round 1

Reviewer 1 Report

In this paper, the authors have developed a PCR assay using an 18S rRNA target to detect very low levels (1 oocyte / g) of Cyclospora cayetanesis infecting berries and causing illness to people who eat the infected foods through the use of nested PCR with 35 cycles for each round. In addition to assay development, phylogenetic analysis and evaluating sensitivity, farm soil and food samples were tested. They have submitted the 18S rRNA sequences obtained in the study into GenBank. The analysis of pathogen source of origin by analyzing SNPs in different haplotypes will be useful in developing future sequence-based assays for source attribution.

The assay could be improved by using melt detection instead of gels as it is more rapid and can be performed in situ following PCR. The authors should test the specificity of the assay against other members of the same genus and family as 18S rRNA genes are well conserved.

The paper is clear and very well-written although there are a few missing articles throughout the paper and minor word edits. Also, the city and state for the company (e.g., Thermo Scientific) only needs stated once at the first use.

Suggested text edits:

Abstract, Line 20: Spell out Cyclospora at the first use

Abstract, Line 23: Add “the” before limit and remove comma after assay

Introduction, Line 41: Add “the” before parasite

Introduction, Line 42: change in farms to “on farms”

Methods, Line 63: Add “the” before first round

Methods, Line 74: Add “the” before PCR

Results, Line 118: Add “the” before first-round and “were” before diluted

Results, Line 119: Change were to “was” before used

Results, Line 175: Delete “It is proposed” that begins the sentence

Results, Line 175: Add “is proposed” before not only

Reviewer 2 Report

The study by Resendiz-Nava et al, developed a nested PCR assay for the detection of the parasite Cyclospora cayetanensis.  The PCR assay targeted the 18S rRNA coding sequences.  The assay was further validated with naturally-contaminated samples of fresh berries and soil samples from berry farms.

This study is addressing a very important issue in food safety and has demonstrated that the pathogen C. cayetanensis can be detected in food and environmental samples.  Unfortunately, the present study did not explain how the current assay compare to what has been already previously published by other groups.  What were other PCR assays used for detecting Cyclospora cayetanensis?  For example, how does this assay compare to the one published 12 year ago by Lalonde et al in Applied and environmental microbiology vol. 74,14 (2008): 4354-8. doi:10.1128/AEM.00032-08.  Moreover, it was unclear to this Reviewer how does the nested PCR documented in the present study compare to what has been previously published by other groups, which were also targeting the 18S rRNA coding sequences (for example citation #29 H.R. Murphy et al., Food Microbiology 69 (2018). 

The current study by Resendiz-Nava et al also conducted a phylogenetic analysis to assess for genetic variation.  The use of haplotype in Figure 5 is inappropriate since the analysis is only determining nucleotide polymorphisms.  There is no analysis of a group of alleles in the offspring that are inherited together from a single parent.  A better term would be to use genotype not haplotype.

How does the current PCR assay compare to other molecular assay currently being developed by other groups targeting the mitochondrial DNA for examining genome sequence variation as a useful marker for assessing genetic heterogeneity among Cyclospora cayetanensis isolates and source-tracking?

Round 2

Reviewer 2 Report

Authors have address comments from this Reviewer.